# OpenReview forum: "Log-Sum-Exponential Estimator for Off-Policy Evaluation and Learning"
_ICLR.cc/2025/Conference — ICLR 2025 Conference Withdrawn Submission_

### Official Review · Reviewer_W6M8 · 2024-10-16

**Soundness:** 4
**Presentation:** 4
**Contribution:** 2
**Rating:** 6
**Confidence:** 3

**Summary:**

This paper introduces the Log-Sum-Exponential (LSE) estimator, a novel approach for off-policy evaluation (OPE) and off-policy learning (OPL) with only logged bandit feedback. The estimator is designed to address high variance, noisy propensity scores, and heavy-tailed reward distributions, common challenges in OPE/OPL. The paper provides theoretical bounds on regret, bias, and variance, and supports the claims with empirical evaluations comparing the LSE estimator against several baseline methods, including truncated importance sampling (IPS) and other state-of-the-art estimators.

**Strengths:**

- The paper studies the practically relevant problem of off-policy learning, particularly for both noisy rewards and propensity scores


- The paper is overall well-written and the arguments are easy to follow


- The paper includes rigorous theoretical analysis of the LSE estimator, providing regret bounds, bias-variance trade-offs, and robustness under noisy rewards. These contributions solidify the estimator’s advantages over existing methods.


- The paper presents comprehensive experiments that demonstrate the estimator’s performance in synthetic and real-world scenarios. The results indicate that the LSE estimator can achieve lower variance and MSE compared to established baselines, supporting the theoretical claims.

**Weaknesses:**

- The paper claims that the proposed method is more robust to noisy rewards and propensity scores, supported by some relevant experiments. However, the analysis could be improved by plotting the policy values of the methods on the y-axis against varying levels of noise in the rewards and propensity scores on the x-axis. This would provide a clearer visual representation of the method's robustness. I consider this a crucial point, as it directly relates to the key advantages of the proposed method.

- The LSE estimator requires parameter tuning (such as the λ parameter), which may complicate practical deployment. The lack of detailed guidance on how to select these parameters could hinder reproducibility and real-world adoption. Additionally, it is unclear how robust the proposed method is to potential errors in setting this parameter.

**Questions:**

Could you provide more practical guidance on how to select the λ parameter? Are there heuristics or automated methods that could assist practitioners in tuning this parameter effectively? How robust the proposed method is to the potential failure in setting the parameter?

---

### Official Review · Reviewer_FxqM · 2024-11-01

**Soundness:** 3
**Presentation:** 3
**Contribution:** 2
**Rating:** 5
**Confidence:** 4

**Summary:**

The paper proposes to use a "log sum exp" estimator for off policy
evaluation and learning in contextual bandits. The main theoretical
claim is that this estimator allows for handling heavy tailed reward
distributions and provides some robustness under distribution shift
(called noisy reward). The main practical claim is simply that the
estimator works better than several baselines in both OPE and OPL
settings.

Questions and Comments:

0. In general, I am unconvinced that heavy-tailed reward is a real
problem. The main reason is that the reward is something that we (the
system builder) gets to design (i.e., if we are building a recommender
system, we decide how to aggregate collected telemetry data into a
scalar reward). It is always possible to design the reward so that it
is bounded at a minimum, which implies that all moments are bounded.

0a. Given this, one possibility is to interpret the heavy-tailed
assumption as a sort of "instance dependence." Suppose that rewards
are bounded in [0,R_max] but that the 1+\eps moment is much much
smaller. Then we'd like statistical guarantees that have a benign
dependence on R_{max}, replacing it with 1+\eps moment. This seems
worthwhile, but if I understand correctly, I think this is already
done in [3] (for the unweighted second moment).

0b. I certainly agree that heavy-tailed importance weight is a real
problem, but this is addressed by a number of other previously
proposed estimators, including IX, clipped importance weights,
smoothing/shrinkage, SWITCH (from [3]), etc.

1. In terms of exposition, it would be helpful to clearly explain how
LSE compares to existing estimators. Table 2 is not very helpful
because the quantities that appear in the bias/variance for LSE do not
appear in the bounds for the others.

1a. As a concrete question: Suppose that rewards were bounded in
[0,Rmax], then we should take \nu_2 = R_{max}^2
P_2(\pi_\theta||\pi_0). Is LSE better than IPS under this setup? It
seems that the positive term of the LSE variance is worse, due to
R_{max}^2 vs R_{max}.

2. Why does the variance bound in Prop 5.7 require bounded second
moment (rather than 1+eps for any eps)? It seems like I can always use
Lemma 5.1 to bound the variance of LSE under bounded (1+eps)
moment. Is the point that Prop 5.7 is tighter due to the
-\lambd\nu_2^{3/2} term? If so, it would be great to make this clear
in the exposition and somehow highlight where/why this refinement
matters. (e.g., maybe it is necessary to address the question in 1a.)

3. The OPE experiment is somewhat unconvincing because the setting is
highly synthetic. It seems that the setup is designed to expose a
weakness of prior methods that is addressed in the present work (i.e.,
heavy tails). But it is not clear that this problem is prevalent in
practice or whether LSE results in some tradeoffs in "more benign" settings.

3a. In particular, it is by now standard to do these experiments on a
wide range of datasets with a broad set of experimental
conditions. See e.g., [1,2]. I strongly recommend that the authors add
such experiments to the paper.

4. Regarding OPE experiments, there are also a number of important missing
baselines, including clipped importance weighting, the SWITCH
estimator of [3], etc.

5. For the OPL experiments, why do we only compare against
unregularized baselines? I think we should view the LSE estimator as a
form of regularization, so it seems natural to also compare with
regularized baselines. In particular, why not compare against MRDR [1]
and DR-Shrinkage [2]? Both can be implemented with a "reward
estimator" that always predicts 0.

6. As alluded to above, the exposition/presentation could be greatly
improved.


Overall, I feel the paper has a decent amount of potential, but isn't
quite there yet. As it stands, it feels a rather shallow. There could
have been a much deeper theoretical and empirical investigation into
the estimator and its properties and this could have made the paper
much better. As mentioned above, I don't think the heavy tailed reward
setting is particularly interesting, but heavy-tailed importance is
definitely a real problem. Focusing the paper on this setting would
have helped, but then this requires a much deeper discussion of prior
work and connections. As a concrete point about this, I believe the
estimator can be formally viewed as a form of regularization, via the
duality between log-sum-exp and entropy regularization. It would have
been nice to consider this view point as a means to connect to other
forms of regularization in the off policy evaluation literature, for
example those developed in [2]. In particular, I think there is
potentially something interesting about the LSE estimator relative to
prior works like [2], namely it is a new form of shrinkage that
explicitly accounts for the n-dimensional nature of the regularization
problem (akin to Stein shrinkage). It would be nice to investigate
this in more detail.


Missing references:

[1] Mehrdad Farajtabar, Yinlam Chow, Mohammad Ghavamzadeh. More Robust
Doubly Robust Off-policy Evaluation. https://arxiv.org/abs/1802.03493

[2] Yi Su, Maria Dimakopoulou, Akshay Krishnamurthy, Miroslav
Dudík. Doubly robust off-policy evaluation with
shrinkage. https://arxiv.org/abs/1907.09623

[3] Yu-Xiang Wang, Alekh Agarwal, Miroslav Dudik. Optimal and Adaptive
Off-policy Evaluation in Contextual
Bandits. https://arxiv.org/abs/1612.01205

Post-rebuttal: I thank the authors for their detailed responses. I have a few comments (below) and I will raise my score to 5. As I mentioned above, I think this paper does have potential but is not quite there yet so I am not quite ready to recommend acceptance.

- I understand the examples provided but I am still unconvinced that heavy-tailed reward is a real problem. We can always make the reward bounded, this is without loss of generality and so the question is which moments are controlled. I appreciate the discussion about 1+eps vs 2+eps moment, and I think the paper would greatly benefit by having this included. More generally, the paper would benefit from a much clearer explanation of (a) the settings that are not handled by prior work and (b) the instance dependent improvements that LSE achieves in settings that are handled by prior work.
- I think LSE _can_ be viewed as regularization, we just need to minimize rather than maximize. Take \lambda < 0, let z be given and define the optimization problem \min_{y} y^\top z + \lambda H(y). Since entropy is concave and \lambda < 0, this is a convex optimization problem. The value at the minimum is exactly LSE_\lambda(z) as defined in Eq (1) of the paper. Note that minimizing makes more sense because we want to be pessimistic (which is standard in offline RL/CB).
- I certainly appreciate the additional experiments and the references to missing related work. Thank you!

**Strengths:**

see above

**Weaknesses:**

see above

**Questions:**

see above

---

### Official Review · Reviewer_PX8q · 2024-11-01

**Soundness:** 3
**Presentation:** 2
**Contribution:** 3
**Rating:** 6
**Confidence:** 4

**Summary:**

The paper proposes a new off-policy estimator based on the log-sum-exp operator, that is motivated by its robustness to heavy tailed rewards. The bias-variance tradeoff of the estimator was studied, its learning guarantees (in terms of regret) as well as its behaviour in reward drift/contaminated reward scenarios. The regret bound in particular has a rate of $O(n^{-\epsilon/(1 + \epsilon)})$ where $n$ the sample size, assuming bounded $(1 + \epsilon)$-th moment ($\epsilon \le 1$). Experiments were conducted to validate the performance of the estimator in specific settings.

**Strengths:**

- The paper tackles the important problem of off-policy estimation.
- The estimator is motivated from a new robustness lens, which is refreshing in OPE/OPL.
- A substantial effort was put to understand the theoretical properties of the estimator.

**Weaknesses:**

The contributions of this work suffer from two major problems that question its validity:

- **Novelty of the estimator**: If the log-sum-exp operator was never used in off-policy estimation, this transform was heavily studied for learning problems just under the "tilted losses" name. See the following work: "On Tilted Losses in Machine Learning: Theory and Applications", Tian Li, Ahmad Beirami, Maziar Sanjabi, Virginia Smith; 24(142):1−79, 2023. (JMLR). The log-sum-exp operator defines their tilted loss in Equation (2) and is studied for learning problems, meaning that OPE/OPL can be seen as a direct application and a lot of teachings from the JMLR paper can be transferred. This work was not cited in the submitted paper and I don't know how the authors can position their paper compared to this one.
**PARTIALLY ADDRESSED IN REBUTTAL.**

- **The convergence rate**: the $O(n^{-\epsilon/(1 + \epsilon)})$ convergence rate seems like a huge error that needs attention. I did not look exactly where this error is coming from, but this rate is unachievable under these conditions. One can easily see it in the case of a one-armed bandit (with $\pi = \pi_0$) and bounded reward $r \in [0, 1]$. These conditions ensure that all the _weighted reward moments_ are bounded and smaller than $1$.  meaning that all $(1 + \epsilon)$ moments are smaller than $1$, ensuring that $\epsilon$ can go to infinity and achieving a convergence rate of $O(n^{-1})$. This cannot be attained (even asymptotically, see CLT) as long as $r$ is not deterministic.
**ADDRESSED IN REBUTTAL.**

Other minor problems can be pointed out as well:
- **Heavy notations**: The structure of the paper and the various definitions/notations used make it hard to follow the proofs, which gets even more opaque in the regret/convergence rate propositions. **PARTIALLY ADDRESSED IN REBUTTAL.** The writing of the paper can be greatly improved.

- **Lack of key baseline**: The logarithmic smoothing estimator that was proposed recently is motivated from the pessimistic lens, and may mitigate the heavy tailed reward problem as it smoothes the _weighted rewards_, contrary to the other baselines used. A comparison against it can strengthen the paper. The paper was cited but not compared to, see "Logarithmic Smoothing for Pessimistic Off-Policy Evaluation, Selection and Learning", Sakhi et. al. 2024
**ADDRESSED IN REBUTTAL.**

**Questions:**

- How can you position your paper compared to "On Tilted Losses in Machine Learning: Theory and Applications"?

   - **PARTIALLY ADDRESSED IN REBUTTAL**: the discussion of "On Tilted Losses in Machine Learning: Theory and Applications" was briefly included and can still be improved.

- Can you explain how the $O(n^{-1})$ convergence rate can be achieved with your method? Can you spot the error?

   - **ADDRESSED IN REBUTTAL**: The results are correct and were clarified during rebuttal.

- Why did not you compare to the Logarithmic Smoothing estimator?

   - **ADDRESSED IN REBUTTAL.** Comparison with the Logarithmic Smoothing estimator was included. The LSE estimator presents favourable performance compared to state of the art approaches.

---

### Official Review · Reviewer_3zEX · 2024-11-04

**Soundness:** 4
**Presentation:** 3
**Contribution:** 3
**Rating:** 6
**Confidence:** 4

**Summary:**

Summary: The authors consider stochastic, contextual bandits where data is collected using a logging policy $\pi_{\log}$. This policy is available for point evaluation. The main idea is to use the LSEA, "log-sum-exponential average" ($f_s(z_1,\dots,z_n) =-1/s \log \sum_{i=1}^n \exp(- s z_i)$, $s>0$) on the top of importance weighted reward data (i.e., estimate the value of policy $\pi$ using $f_s(Z_1,\dots,Z_n)$ where $Z_i = R_i \pi(A_i|X_i)/\pi_{\log}(A_i|X_i)$, $X_i$ is context for the $i$th data point, $A_i$ is the corresponding action logged, and $R_i$ is the associated reward). The authors show generalization bounds on how well this estimator approximates the true value of policies, as well as bounds on the (simple) regret of the policy which, in a given class, gives the best LSEA. The bounds depend on the smoothing parameter $s$ (the authors use $\lambda = -s$) and $\epsilon>0$, which is a constant such that $\mathbb{E}[ |Z_i]^{1+\epsilon} ]\le \nu$ (since $Z_i$ depends on the policy, this is demanded to hold uniformly for all policies). We shall call this a moment constraint on the importance weighted reward. In particular, it is claimed that the "rate" of the suboptimality of the chosen policy is $O(n^{-\epsilon/(1+\epsilon)})$. In addition to the theoretical result, the authors also show empirical evidence that their approach is a good one. For this, two setting are considered: a synthetic problem (gaussian policies, etc.), and another problem based on transforming EMNIST (and also FMNIST) to a bandit problem.

Significance/novelty: The problem is of high significance, "fixing importance weighting estimators" is also of high importance. Bringing LSEA to this problem is a new and nice idea. The theoretical results are of interest, as well, and the empirical evidence presented is promising. The robustness result is OK, but I could not see its significance (yes, we can do this.. until I see a lower bound, it is unclear how lose this or similar results are).

Soundness: I believe the results are correct, although I did not verify the details of the proofs. However, both the results and the techniques look reasonable (see a few questions below). I wished that Table 3 comes with error bars, and/or some presentation of the distributional properties of the various estimators, especially, given the emphasis on "heavy tailedness".

Novelty/related work: The authors motivate using LSEA by arguing that the data will be "heavy tailed". This is fine, but then I don't see the relevant literature on estimating means with heavy tail data discussed in the paper. The discussion of the rest of the literature is fine. This literature is big, so we should not expect a comprehensive discussion. I found it awkward that while the 2024 paper of Sakhi et al., which introduced the logarithmic smoothing estimator, is cited, it is not compared to and the idea of logarithmic smoothing is not discussed, even though it feels also related to the present paper in that I expect it also perform well in the heavy-tailed setting of the paper (the logarithmic transformation of the data, intuitively, turns large tails into smaller ones).

Presentation: I found the presentation to be fine.

Some minor issues:
Is the number of actions finite? If not, what is $\pi(a|x)$? Density of distribution over arms with respect to some base measure?

**Strengths:**

1. log-sum-exp was not studied in this context and it seems it should be studied
2. the results obtained are reasonable; there is some nice math (buried in the appendix)
3. the paper looks at both theory (minimum guarantees) and also garners some empirical evidence in favour of the method.

**Weaknesses:**

The paper feels a bit undercooked. The upper bounds are a little messy and little to no effort is spent on explaining them. The relation to the heavy-tailed mean estimation literature should have been discussed. It is unclear how the results fair compared to the recent work on logarithmic smoothing.

**Questions:**

For $\epsilon$ large enough, the rate is better than $1/\sqrt{n}$. How can this be true? What is the mechanism that allows us to get a rate better than the statistical rate? (In finite armed bandit, for sure, asymptotic rates will be exponential, but if the result has this asymptotic flavor, I am not going to be too excited about it, because in a contextual setting I don't think that the asymptotics has any chance of "kicking in").

In essence, the authors point out that importance weighted rewards are heavy tailed; hence the focus in the analysis on moment constraints. This raises two questions: Why not consider the big literature on estimating the mean of heavy tailed distribution? Why pick LSEA? Has LSEA been analyzed in that literature? I expected the authors to at least look at this question in the paper. Secondly, with moment conditions only, I expect weaker results. For example, in the literature on mean estimation with heavy tail iid data, we know that the "subgaussian like tail bounds" work for "fixed delta" (the error probability where the tail bound holds is an input to the algorithm and this weakness is inherent to the problem not the algorithms). I suppose we are paying a similar price here. But this is not very clear from the presentation and I would have expected an honest discussion of this (I am guessing this limitation comes from that the results only hold for $n$, the sample size, "large enough"). The bigger question then is: Alternative estimators avoid this problem. So is LSEA inferior to those in this sense?

---

### Author Response · Authors · 2024-11-30
**Rebuttal Summary (Experiments)**

# Rebuttal Summary

We are grateful to all reviewers for their insightful comments and feedback. In this post, we summarize our both the original experiments from our work and those additional experiments we conducted during the rebuttal period to address reviewer questions and suggestions.

# Experiments

## OPL Experiments:

- **SupervisedToBandit** (Section 6.2 and Section G.2): These experiments are based on the supervised-to-bandit experimental setup explained. Detailed experiments including normal, noisy propensity scores, and noisy rewards settings on FMNIST and EMNIST datasets for evaluating different estimators are reported. *LS-LIN* and *OS* estimators were added during the ***rebuttal*** period (`reviewers PX8q, FxqM and 3zEX`).
-  **Model-based estimator** (Section G.3): The estimator based on Double-Robust estimation are evaluated and compared. *DR-SWITCH*, *DR-SWITCH-LSE*, *DR-OS* and *MRDR* estimators were added  during the ***rebuttal*** period (`reviewer FxqM`) .
- **Real-world Experiments** (Section G.4) : The real-world experiments evaluate model performance on an original bandit dataset collected in a real-world circumstance. KuaiRec dataset results are reported. *LS-LIN* and *OS* estimators were added during the ***rebuttal*** period (`reviewers PX8q, FxqM and 3zEX`).
- **Sample size $n$ effect** (Section G.5): Analyzes how the sample size impacts the performance of different estimators.
- **Effect of hyperparameter $\\lambda$** (Section G.6): explores the effect of the hyperparameter $\\lambda$ on the LSE estimator.
- **$\\lambda$ selection for OPL** (Section G.7): **$\\lambda$** selection based on number of samples is compared to grid-search selection.

## OPE Experiments:

- **Synthetic Experiments** (Section 6.1 and G.1): Synthetic datasets, Gaussian and Lomax, are generated to compare different estimators based on their ability to estimate target policy's average reward. *LS*, *LS-LIN*, and *OS* estimators were added during the ***rebuttal*** period (`reviewers PX8q, FxqM and 3zEX`).
- **$\\lambda$ selection for OPE** (Section G.7.2-G.7.3, **rebuttal**, `reviewer W6M8`): Proposes **$\\lambda$** value for OPE and compares LSE with this selection against other estimators.
- **Sensitivity to $\\lambda$** (Section G.7.4, **rebuttal**, `reviewer W6M8`): Shows estimator sensitivity to hyperparameter selection.
- **OPE with noise** (Section G.8, **rebuttal**, `reviewer W6M8`): Discusses robustness of different estimators to noise.
- **Distributional behavior in OPE** (Section G.9, **rebuttal**, `reviewer 3zEX`): Investigates error distribution of different estimators.
- **Comprehensive comparison with LS** (Section G.10, **rebuttal**, `reviewers PX8q and 3zEX`): Detailed experiments comparing LS and LSE estimator sensitivity to hyperparameter selection.
- **OPE on real-world datasets** (Section G.11, **rebuttal**, `reviewer FxqM`): Experiments on real classification datasets from UCI repository.

### We would be happy to conduct additional experiments to address any remaining questions or concerns before the end of discussion period.

---

> ### Author Response · Authors · 2024-11-30
> **Rebuttal Summary ( Theoretical Results and Discussions)**
>
> # Rebuttal Summary
>
> Following [this post](https://openreview.net/forum?id=89EjtiGWVS&noteId=vIoMn6HDxZ),  we summarize our both the original theoretical results from our work and those additional theoretical results and discussions we added during the rebuttal period to address reviewer questions and suggestions.
>
> ## Theoretical Results and Discussions
>
> - **LSE estimator and KL Regularization** (Section C.1, **Rebuttal**, `reviewer FxqM`): We explore the connection between the LSE estimator with a negative parameter and KL regularization.
> - **Bias and Variance** (Section 5.3): The bias and variance of the LSE estimator are analyzed. The upper bound on Variance is improved during **rebuttal** and shown that the LSE variance is less than the IPS estimator (`reviewer PX8q`).
> - **Bias and Variance Comparison** (Section D.1.1): Updates to Table 6 provide a comprehensive comparison of bias and variance across all estimators, including the addition of comparisons for the *OS* and *LS* estimators during the ***rebuttal*** period (`reviewers FxqM, PX8q and 3zEX`).
> - **Regret analysis** (Section 5.2): The regret upper bound under *heavy-tailed assumption* is provided.
> - **Implicit Shrinkage** ( Section D.7, **Rebuttal**,`reviewer FxqM`): The implicit shrinkage property of the LSE estimator is investigated.
> - **Sub-Gaussian Bound** (Section D.6, **Rebuttal**,`reviewer 3zEX`): The sub-Gaussian upper bound on the absolute generalization error (concentration inequality) is provided.
> - **Noisy reward** (Section 5.4): The robustness of the LSE estimator under noisy reward settings is analyzed.
> - **Estimated propensity scores** (Section E): The robustness of the LSE estimator under estimated propensity scores is examined.
> - **PAC-Bayesian Discussion** (Section D.5): A PAC-Bayesian version of our regret bound is derived, offering additional theoretical insights.
> - **Comparison with other estimators (Section 5):** Table 2 presents a comprehensive comparison of the LSE estimator against all baseline methods. *OS* and *LS* were added during the ***rebuttal*** period (`reviewers FxqM, PX8q and 3zEX`).
> - **Detailed comparison with tilted empirical risk** (Section D.1.8, **Rebuttal**,`reviewer PX8q`): A detailed comparison with tilted empirical risk framework is provided.
> - **Comparison under Bounded Reward**  (Section D.1.7,**Rebuttal**,`reviewer FxqM`): A full comparison of other estimators with LSE under bounded reward assumption is provided.
> - **Comparison with Specific Estimators:**
>   - **PM Estimator** (Section D.1.2): A detailed theoretical comparison with the PM estimator.
>   - **ES Estimator** (Section D.1.3): A detailed theoretical comparison with the ES estimator.
>   - **IX Estimator** (Section D.1.4): A detailed theoretical comparison with the IX estimator.
>   - **LS Estimator** (Section D.1.5, **Rebuttal**,`reviewers PX8q and 3zEX`): A detailed theoretical comparison with the LS estimator.
>   - **OS Estimator** (Section D.1.6, **Rebuttal**,`reviewer FxqM`): A detailed theoretical comparison with the OS estimator.
> - **Comparison with Assumption 1 in Switch Estimator** (Section D.1.9, **Rebuttal**,`reviewer FxqM`): We provide a detailed comparison between our heavy-tailed assumption and Assumption 1 from the switch estimator framework.
> - **Additional Related Work** (Section A, **Rebuttal**, `reviewers FxqM and 3zEX`): We have expanded the related work to include works on *generalization error under heavy-tailed assumptions*, *mean estimation under heavy-tailed distributions*, and *heavy-tailed rewards in bandits* and *reinforcement learning*.
>
> ## We would be happy to conduct additional theoretical result or discussion to address any remaining questions or concerns before the end of rebuttal.

---

### Note · Authors · 2025-01-22

I have read and agree with the venue's withdrawal policy on behalf of myself and my co-authors.